# Influence of Material Removal Strategy on Machining Deformation of Aluminum Plates with Asymmetric Residual Stresses

**DOI:** 10.3390/ma16052033

**Published:** 2023-03-01

**Authors:** Yang Li, Ya-Nan Li, Xi-Wu Li, Kai Zhu, Yong-An Zhang, Zhi-Hui Li, Hong-Wei Yan, Kai Wen

**Affiliations:** 1State Key Laboratory of Nonferrous Metals and Processes, GRINM Group Co., Ltd., Beijing 100088, China; 2GRIMAT Engineering Institute Co., Ltd., Beijing 101407, China; 3General Research Institute for Nonferrous Metals, Beijing 100088, China

**Keywords:** aluminum alloy, material removal strategy, machining deformation, finite element simulation, asymmetric residual stress

## Abstract

In this paper, the effects of material removal strategies and initial stress states on the machining deformation of aluminum alloy plates were investigated through a combination of finite element simulation and experiments. We developed different machining strategies described by Tm+Bn, which removal m mm materials form top and n mm materials from the bottom of the plate. The results demonstrate that the maximum deformation of structural components with the T10+B0 machining strategy could reach 1.94 mm, whereas with the T3+B7 machining strategy was only 0.065 mm, decreasing by more than 95%. The asymmetric initial stress state had a significant impact on the machining deformation of the thick plate. The machined deformation of thick plates increased with the increase in the initial stress state. The concavity of the thick plates changed with the T3+B7 machining strategy due to the asymmetry of the stress level. The deformation of frame parts was smaller when the frame opening was facing the high-stress level surface during machining than when it was facing the low-stress level. Moreover, the modeling results for the stress state and machining deformation were accurate and in good accordance with the experimental findings.

## 1. Introduction

Aluminum alloy is commonly used as a crucial structural material, notably in the aerospace industry, because of its improved processability, high specific strength, and great filling capacity [1,2,3]. In order to develop high-speed and heavy-load aircraft, the main products of aluminum alloy applied in the aerospace field are currently large monolithic components made of aluminum alloy thick plates [4]. However, monolithic structural components tend to experience excessively high deformation or even cracking in the machining process, which can significantly hinder their development and use [5,6,7].

The problem of distortion in the machining of thick plates mainly arises from three aspects. The initial residual stress state substantially influences the distortion during the machining process [7,8,9,10,11,12]. Heat-treatable aluminum alloys are subjected to solution-quenching heat treatment to achieve their required performance. During the quenching treatment, severe temperature gradients are created between the surface and the interior, resulting in high levels of residual stress [13]. The high material removal rate (typically more than 90%) during the machining process results in the release and redistribution of residual stress, even though the stress level inside the thick plate can be effectively reduced through pre-processing techniques such as pre-stretching and pre-compression [14,15,16]. There has been a lot of research done currently on the relation between machining deformation as well as the initial stress state. The impact of the initial stress state on the machining deformation of gearboxes was researched by Husson et al. [11]. The outcomes showed that heat treatment may be used to control residual stress through the semi-finished product and can lessen machining deformation. According to Fu et al. [12], the initial residual stress had a significant impact on forecasting and managing the machining deformation. They described two piecewise calculation techniques with reasonably good prediction accuracy, and finite element modeling and machining tests were used to confirm the two new techniques’ correctness.

The deformation of the material may be influenced by the cutting circumstances, including the cutting speed, feed rate, and depth of cut [17,18,19,20]. During the milling process, the back-tool face and cutter tooth extrusion may result in plastic deformation on the machined surface [21]. The plastic deformation may prevent the adjacent component’s neighboring portion from recovering from its deformation, leaving residual stress on the part’s surface. The tool’s rapid rotation also acts as a heat source of motion on the substrate surface of the workpiece, which can disperse the initial residual stress states [20]. Regarding the second issue, the following has been covered in the scientific literature on machining deformation simulation and analyzation. Through a combination of simulation and experiments, Wu et al. [18] examined the machining distortion of aerospace monolithic parts and carried out enhancement of the cutting parameters. Li et al. [19] examined the impact of the depth of cut on the revolution of stress during the machining process.

The material removal strategy also affects the distortion of thick plates during the machining process [14,22,23]. Different material removal strategies can change the sequence and manner of residual stress release, so the machining distortion of thick plates can be reduced with a suitable material removal strategy.

It is clear from the literature that much research has been dedicated to understanding the impact of initial residual stress state on machining deformation in the milling of monolithic aircraft components and finding ways to minimize the final distortion. However, residual stress in blank materials is often asymmetrical in practical production processes, and few studies have examined the impact of this asymmetry on thick plate machining distortion. There is also a lack of research on machining strategies for specific types of frame-type structural components.

In this paper, asymmetric distribution of residual stress of the thick plates were implemented by means of finite element simulation and asymmetric spraying experiments. It was investigated how the internal stress of the thick plate varied under various spray settings. Based on this, it was addressed how the machining strategy, asymmetrical residual stress states, and their linkage affected the machining deformation of thick plates. It was the first time to clarify how the asymmetrical stress states influenced the deformation of the plate during the machining process, and we also observed how frame orientations affected the way that plates with asymmetric stress states deformed during machining. It could provide certain support for processing deformation control of the aviation aluminum alloy.

## 2. Materials and Methods

### 2.1. Simulation Section

#### 2.1.1. Assumption of Numerical Simulation

To facilitate the analysis and modeling of the complex heat transfer process of aluminum alloy spray cooling, and analyze the stress distribution and revolution during spray cooling more effectively and quickly, the following presumptions were made: (1) the thick plate was assumed to be an isotropic material with uniform internal components and tissues and no texture; (2) it was assumed that the initial stress of the plate during spray cooling was zero because the plate underwent a lengthy high-temperature solution treatment that fully relieved the internal residual stress at high temperatures; and (3) as the goal of the solution process of aluminum alloys was to completely dissolve precipitates and improve the material performance, the thick plate was assumed to have little second-phase precipitation during spray cooling, and the latent heat of the phase change during this process was thus assumed to be zero [24].

#### 2.1.2. Geometric Model and Material Parameters of Numerical Simulation

The dimension of the finite element model was 320 mm × 60 mm × 26 mm. The size of the element was 2 mm × 2 mm × 1 mm and the total number of elements was 124,800. Table 1 displays the density (ρ), thermal conductivity (λ), specific heat capacity (c), linear expansion coefficient (α), yield strength (Rp0.2), and elastic modulus (E) of the 7055 alloy utilized in the simulation. The quenching temperature field’s starting point was set at 475 °C, and the final temperature was placed at 25 °C.

In the numerical modeling of quenching, the heat transfer coefficient is an essential boundary condition. According to earlier studies [25,26,27], the evolution of temperature fields in plates during quenching may be precisely described by employing interfacial heat transfer coefficients computed by inverse methods. In order to determine temperature fields throughout the quenching process, the average heat transfer coefficient was used as the boundary condition. Table 2 displays the average heat transfer coefficient values for various water flow rates used for spray quenching. The flow rate of each nozzle was measured in real-time using a turbine flowmeter. The maximum flow rate of a single nozzle was found to be 0.64 m^3^/h. The flow rates of the single nozzles were then decreased in sequence to 0.48 m^3^/h, 0.32 m^3^/h, and 0.16 m^3^/h, and subsequent spraying tests were conducted. For the ease of description, the spraying flow rates of different single nozzles were calibrated as 1, 0.75, 0.5, and 0.25 in sequence.

The ANSYS sequential coupling simulation method was used to solve the temperature field results of the quenching process through superheated convection. The element type used for temperature field simulation was solid70. The temperature field results were then introduced into the stress field analysis. To showcase the impact of different material removal strategies on frame monolithic components, the machining simulation approach known as “birth and death” was used [10,28]. The typical structural component employed in this research is depicted in Figure 1. The 320 mm × 60 mm × 16 mm machined product had a 2 mm wall thickness. The element type used for stress field simulation was solid45.

### 2.2. Experimental Section

#### 2.2.1. Experimental Materials

Experiments were performed on thick plates of the 7055 aluminum alloy with the same dimension as the finite element model. It is well known that surface roughness can impact heat transfer during quenching, leading to uneven residual stress distribution [29]. To address this, the sprayed surfaces of the specimens were ground to a consistent roughness of 0.8 μm.

#### 2.2.2. Spray Quenching Experiment

The spray quenching equipment included heating devices, spray systems, and water circulation systems, as shown in Figure 2. The water circulation system includes a water tank, a centrifugal pump, valves, filters, flow meters, and a stainless-steel hose. The water temperature in the circulation pool was maintained at 25 ± 1 °C. The spray pressure and flow rate were adjusted by the frequency conversion centrifugal pump (PRODN CHM4-2DC) and were calibrated with the flow meter. First, the sample was treated with a solid solution furnace at 475 °C × 4 h to eliminate the residual stress before quenching, while ensuring that the temperature inside the sample was consistent. Then, the sample was quickly transferred to the spray device and the water pump valve was opened. The sample transfer time was controlled within 10 s, and the quenching time was not less than 120 s.

#### 2.2.3. Characterization Method of Residual Stress and Deformation

The residual stress on the surface of the sample was measured by the X-ray diffraction method of the μ-X360s X-ray stress analyzer (Figure 3a). The deformation of the frame structural parts was measured using a 3D coordinate machine (EXPLORER 06.08.06, HEXAGON, Stockholm, Sweden, Figure 3b).

## 3. Results and Discussions

### 3.1. Residual Stress Distribution of the Asymmetrical Spray Quenching

The finite element simulation of the asymmetric spray involved fixing the spray flow on the top surface of the thick plate to 1, and the spray flow on the lower surface changed from 1 to 0.25. After asymmetric spray quenching of the thick plate, the stress state remained “compressive on the surface and tensile in the core”, as shown in Figure 4. The ultimate surface stress was inconsistent, nevertheless, as a result of the divergent spray flow rates on the upper and lower sides of the plate. Significantly more stress was present on the side with a high spray flow rate compared with the side with a low spray flow rate. The overall stress of the thick plate dropped as the spray flow rate was decreased. As shown in Figure 5, the maximum surface compressive stress on the thick plate with a high spray flow rate decreased from 150.8 MPa to 123.9 MPa when the spray water flow rate on one side was reduced from 1 to 0.25, while the maximum surface compressive stress on the side with a low spray flow rate decreased from 150.8 MPa to 106.3 MPa. Meanwhile, the maximum core tensile stress of the thick plate was reduced from 83.2 MPa to 59.6 MPa.

In order to verify the accuracy of the simulation results, experiments were conducted on 7055 aluminum alloy thick plates with the same conditions as the simulation model. The surface stresses of the thick plates were measured. Figure 6 illustrates the schematic diagram of the residual stress characterization path.

Figure 7 shows a comparison of the surface stress distribution between the simulation and experiment. The surface stress data obtained from the simulation were more uniform, while the measured data were more volatile and discrete when compared with the simulated results. This discrepancy can be attributed to factors such as the surface quality and grain size uniformity of the thick plate. The maximum error of the surface residual stress between the simulation and experiment was within 10MPa. Meanwhile, the overall volatility was within an acceptable range, which validated the accuracy of the model.

After cooling the thick plate, the stress level in the thickness direction exhibited a parabolic distribution. The compressive stress gradually decreased from the surface to the core, and the stress value reached zero at about 1/4 of the plate thickness. The stress state progressively shifted from compressive to tensile and increased, reaching the maximum tensile stress at around half of the plate thickness. As shown in Figure 8, with the gradual reduction in spray flow on one side, the surface stress on both sides was reduced, but the change in the low spray flow surface was greater than the high spray flow surface. As the difference between the two sides of the spray flow increased, the high spray flow surface eventually absorbed the greatest amount of tension that was in the thick plate’s core.

### 3.2. Residual Stress and Deformation under Different Machining Strategies and Initial Stress States

#### 3.2.1. Model and Material Removal Strategy

The frame part had a size of 320 mm × 60 mm × 16 mm with a wall thickness of 2 mm and the size of the initial thick plate was 320 mm × 60 mm × 26 mm. To this end, 11 material removal strategies were designed, as illustrated in Figure 9. The thick plate needed to remove 10 mm of material in the thickness direction with 1mm as a layer. Thus, for the strategy of Tm+Bn (m + n = 10, m = 0 ~ 10), m layers were removed from the top of the thick plate, while n layers were removed from the bottom.

#### 3.2.2. Effect of Machining Strategies on the Distribution of Residual Stress and Deformation

Figure 10 shows the stress state of the frame parts along the length direction (S11 direction) after machining the thick plate. As can be observed, the compressive stress along the S11 direction under different machining strategies was mainly concentrated at the top of the frame wall. The stress level was small and relatively uniform, and the stress concentration was primarily located at the point of contact between the reinforcement and the bottom of the frame part. This was due to the gradual release of the original residual stress inside the plate as the material was removed, which was then concentrated at the top of the frame wall of the frame part. To balance the compressive stress at the reinforcement, large tensile stress was generated at the front and rear sides of the plate.

From the change in the stress magnitude of the frame parts under different processing strategies, as shown in Figure 11, the top compressive stress of the frame wall showed a trend of decreasing and then increasing as more material was removed from the top. The maximum compressive stress at the top of the frame wall was 54.8 MPa when no material was removed from the top and it decreased as more material was removed. The maximum compressive stress at the top of the frame wall was reduced to 35.5 MPa when 4 mm of material was removed from the top and increased as more material was removed. The maximum compressive stress at the top of the frame wall increased to 41.9 MPa when 10 mm was removed from the top.

The tensile stress in the frame wall of the frame part increased initially and then decreased as more material was removed from the top. The maximum tensile stress in the frame wall of the frame part was 27.8 MPa when no material was removed from the top, and gradually increased as more material was removed. The maximum tensile stress in the frame wall of the frame part increased to 31.4 MPa when 4 mm of material was removed from the top, and then gradually decreased as more material was removed. The maximum tensile stress in the frame wall of the frame part decreased to 23.6 MPa when 10 mm of material was removed from the top.

Figure 12 illustrates the cloud maps of deformation in the frame part, with a magnification of ten times, along the thickness direction for the thick plate under different machining strategies. The displacement field clouds along the thickness direction demonstrated that the overall deformation pattern of the thick plate changed as a result of different machining strategies. As more material was removed from the top of the plate, the deformation changed from a “convex” shape to a “concave” shape. Meanwhile, the characterized path of the deformation at the bottom of the frame part is shown in Figure 12.

Figure 13 illustrates the deformation results of the bottom of the frame parts under different machining strategies (Path_A). It is evident that the overall deformation tended to decrease before increasing as the material removed from the top gradually increased. The deformation form of the thick plate changed from “convex” to “concave”. When 10 mm of material was removed from the top, the thick plate could bend up to 1.94 mm. However, when 3 mm of material was removed from the top, the overall deformation was within 0.1 mm, a decrease of roughly 95%, and the deformation of the thick plate was minimized. This shows that following the machining process, plate deformation could be greatly reduced with the use of an appropriate machining approach. Furthermore, it can be inferred from the outcomes of the stress field investigation that the plate’s overall stress level was quite low after removing 3 mm of material from the top. Therefore, for the investigated structural section, the top 3 mm of material removal was the best machining strategy.

Three machining techniques, T0+B10, T3+B7, and T10+B0, were chosen for machining experiments on the CNC milling machine to confirm the accuracy of the simulation results. All of the cutting parameters remained constant, as shown in Table 3, to illustrate the impact of material removal strategies on the machining deformation of plates. Photos of the machining operations and specimens using various machining strategies that were positioned on the same platform are shown in Figure 14. The overall deformation pattern of the plate changed under various machining procedures, as seen from Figure 14b, which was consistent with the pattern of simulation results. Figure 15 displays the outcomes of the machined samples’ 3D characterization. Under the three machining strategies, the simulation produced maximum deformation values of 0.906 mm, 0.065 mm, and 1.94 mm, whereas the experiments produced maximum deformation values of 1.017 mm, 0.303 mm, and 2.065 mm. The variation was 0.111 mm, 0.238 mm, and 0.125 mm, respectively. There are several variables that could have an impact on the actual deformation during machining, but the T3+B7 group’s deviation was relatively high, presumably as a result of the idealized simulation settings. In conclusion, the simulation’s results were in strong agreement with the outcomes of the experiments, proving the simulation’s validity.

#### 3.2.3. Coupling Effect of Machining Strategies and Initial Stress States on the Distribution of Machining Deformation

A simulation was carried out to investigate 11 machining strategies for each of the four asymmetric sprayed plates. The deformation along the thickness direction of frame-type parts after machining is shown in Figure 16 for plates with different initial stress states. The lower the overall stress level of the sheet, the smaller the final deformation. Comparing the two sets of data with a spraying ratio of 1:1 and 1:0.25, the maximum stress of the plate was reduced from 150.8 MPa to 123.9 MPa, a reduction of approximately 17.8%; the deformation was reduced from 2.42 mm to 1.72 mm, a reduction of approximately 28.9%. This implies that reducing the total stress level can successfully lessen the plate’s distortion following machining. However, because of the asymmetry of the stress level, the concavity will vary as a result of the partial machining strategy’s stress reduction on one side.

Under most machining strategies, the overall deformation pattern of the thick plate does not change significantly with decreasing stress levels. However, for certain machining strategies, the concavity of the thick plate changed as the asymmetry of the stress level increased, as shown in Figure 17. Under the T3+B7 machining strategy, the deformation at the bottom of the frame changes from a “convex” shape at the stress level of 1:1 to a “concave” shape at the stress level of 1:0.25. This may be due to the change in the stress release pattern during the processing of thick plates caused by the asymmetry of stress levels.

To examine the impact of stress state asymmetry on the machining deformation of thick plates, two machining techniques were developed for plates with asymmetric stress levels in the thickness direction, as shown in Figure 18, in which strategy_A represents that the frame-mouth was adjacent to the surface with a higher stress level, while strategy_B represents that the frame-mouth was adjacent to the surface with a lower stress level. As shown in Figure 19, the final deformation of the parts varied depending on the orientation of the frame-mouth and the stress level. As the difference in stress level increased, the difference in deformation between the two strategies became more pronounced. When the frame-mouth was oriented towards the high-stress level, the deformation of the part was significantly smaller than that towards the low-stress level under the different initial stress states. Furthermore, when the stress level changed from 1:0.75 to 1:0.25, the overall deformation of the parts became smaller, but the difference between the different machining strategies was greater. These results highlight the importance of considering stress level asymmetry and frame-mouth orientation when designing machining strategies for thick plates.

## 4. Conclusions

(1)With the reduction in the spray flow rate following spray quenching, the stress level in thick aluminum alloy plates also decreased. As the difference in spray flow rate between the two sides steadily widened during asymmetric spray quenching, the stress level in the core of the thick plate gradually migrated toward the high spray flow surface. The simulation model and the starting stress states were accurate, as shown by the good agreement between the simulation and experimental data.(2)The final machining distortion of the thick plate could be significantly decreased by using an appropriate material removal approach. The maximum deformation could be reduced by 95% when the top 3 mm of material was removed. Moreover, the deformation results of the machining experiment proved the accuracy of the simulation results.(3)Under the same machining strategy, the lower the stress level of the plate, the smaller the final deformation. Under the T3+B7 machining strategy, the deformation at the bottom of the frame changed from a “convex” shape to a “concave” shape because of the asymmetry of stress levels. Meanwhile, the deformation of the part was significantly smaller when the frame-mouth was oriented towards the high-stress level.

## Figures and Tables

**Figure 1 materials-16-02033-f001:**
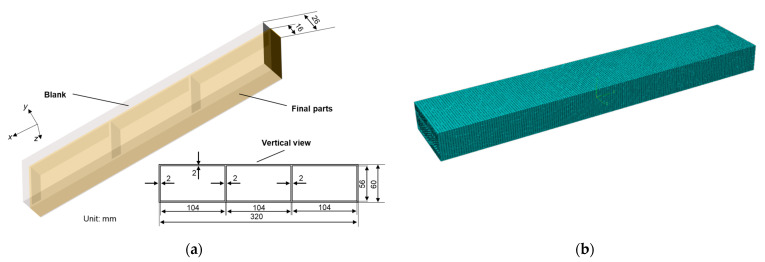
Diagram of finite element model: (**a**) dimension of model and (**b**) mesh part.

**Figure 2 materials-16-02033-f002:**
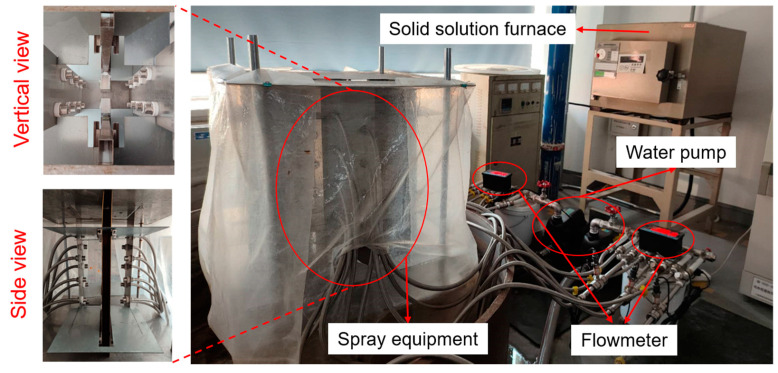
Spray quenching equipment.

**Figure 3 materials-16-02033-f003:**
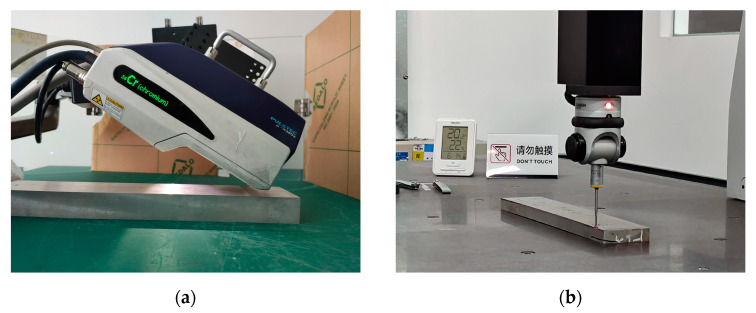
Schematic diagram of the measurements: (**a**) surface stress and (**b**) deformation.

**Figure 4 materials-16-02033-f004:**
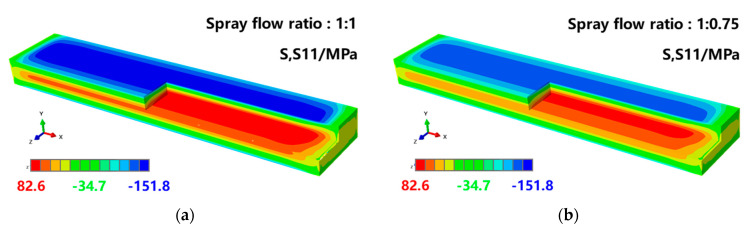
Cloud maps of residual stress in thick plates under symmetric spray quenching: (**a**) 1:1; (**b**) 0.75:0.75; (**c**) 0.5:0.5; (**d**) 0.25:0.25.

**Figure 5 materials-16-02033-f005:**
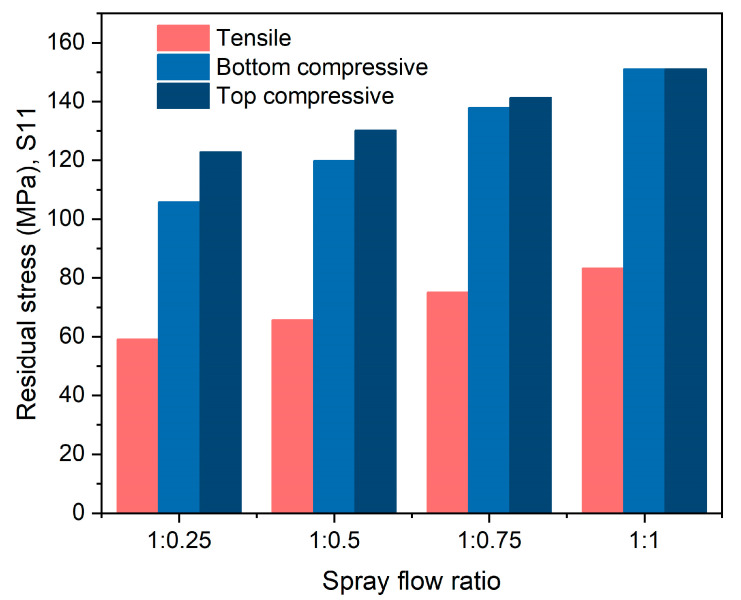
Maps of residual stress with asymmetric spray quenching.

**Figure 6 materials-16-02033-f006:**
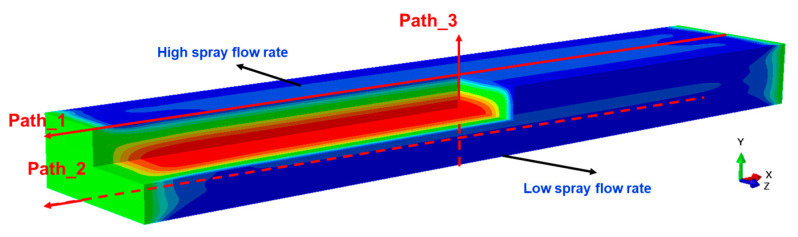
Schematic diagram of the residual stress characterization path.

**Figure 7 materials-16-02033-f007:**
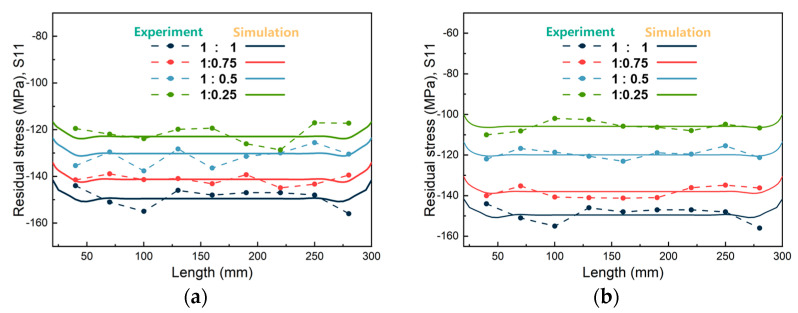
Comparison of residual stress characterized by simulation and experiment along different paths: (**a**) Path_1; (**b**) Path_2.

**Figure 8 materials-16-02033-f008:**
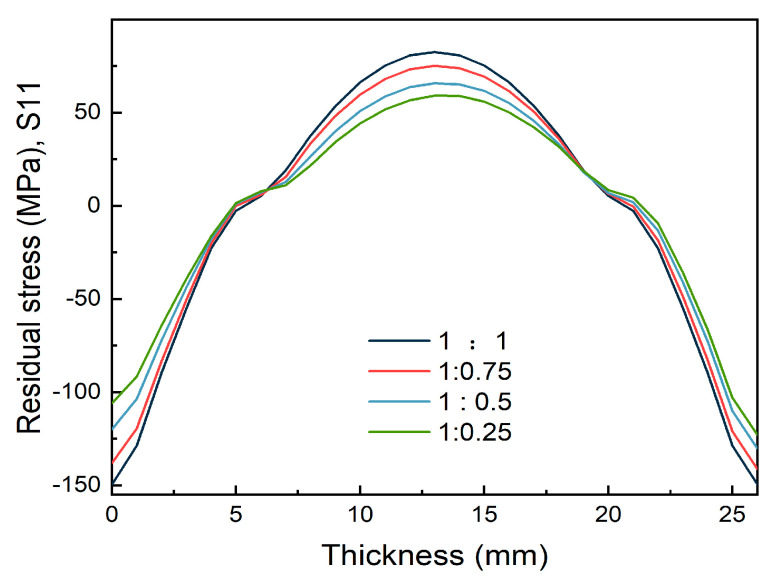
Internal stress of thick plates along Path_3.

**Figure 9 materials-16-02033-f009:**
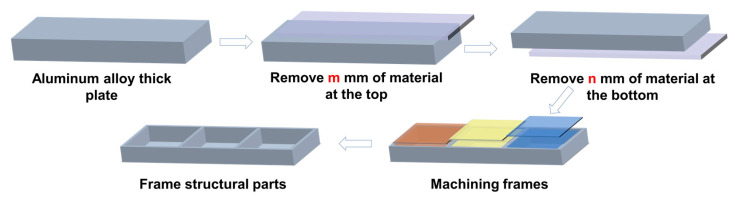
Diagram of the material removal strategy.

**Figure 10 materials-16-02033-f010:**
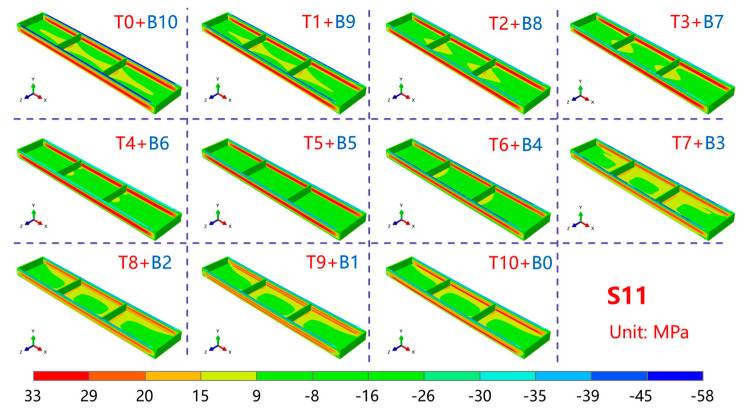
Maps of residual stress in components with various machining strategies.

**Figure 11 materials-16-02033-f011:**
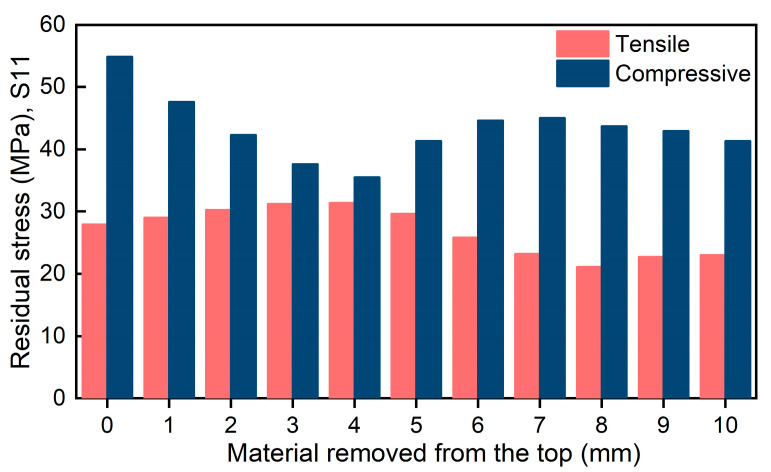
Comparison of maximum stress under different machining strategies.

**Figure 12 materials-16-02033-f012:**
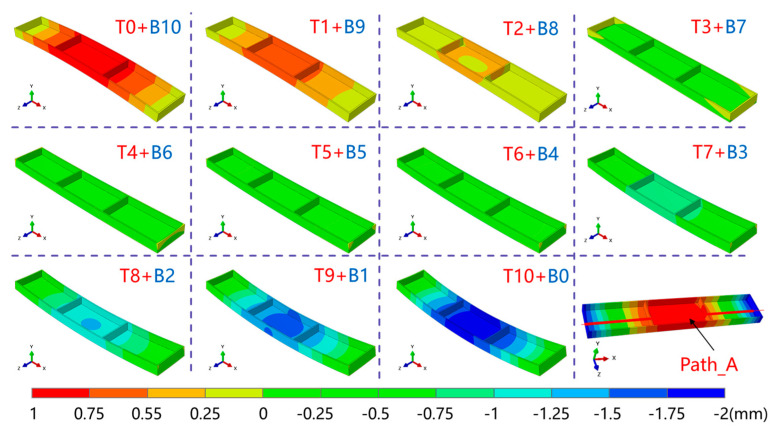
Cloud maps of deformation in parts under different machining strategies and the deformation characterization path.

**Figure 13 materials-16-02033-f013:**
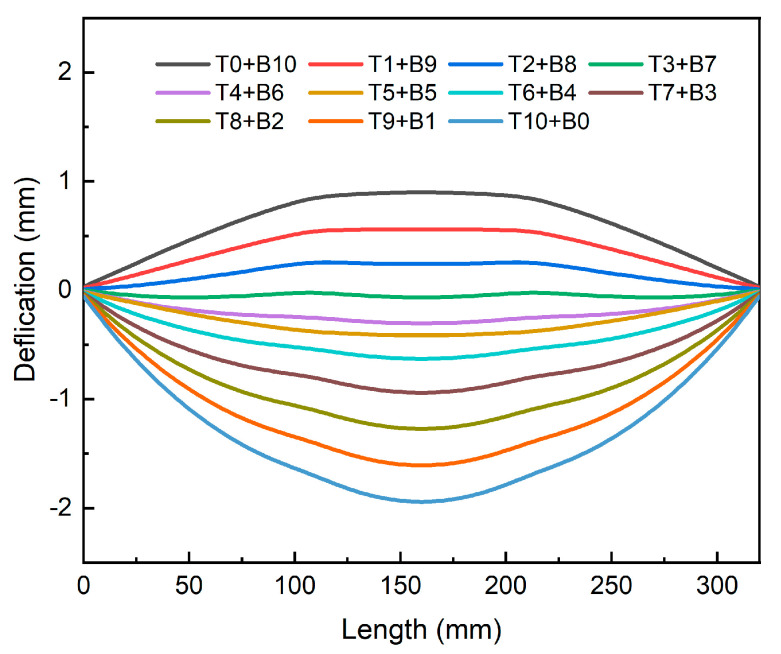
The bottom deformation of parts under different machining strategies along Path_A.

**Figure 14 materials-16-02033-f014:**
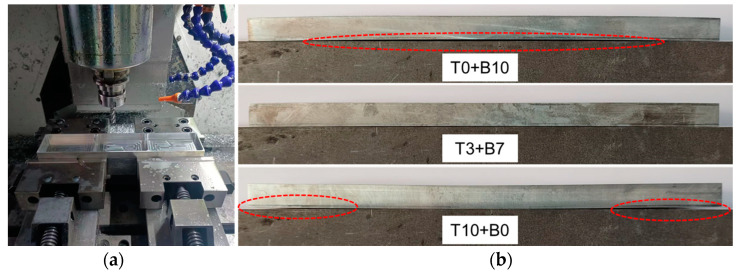
Photos of (**a**) the machining process and (**b**) different specimens put on the same platform.

**Figure 15 materials-16-02033-f015:**
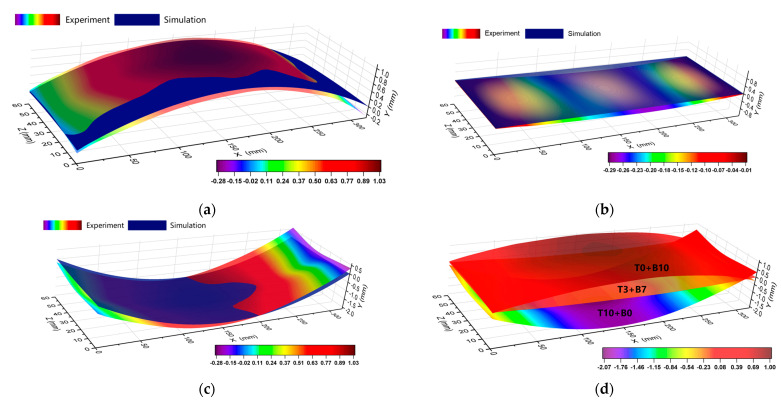
Comparison of experiments and simulation under different machining strategies: (**a**) T0+B10; (**b**) T3+B7; (**c**) T10+B0; (**d**) comparison of the three strategies.

**Figure 16 materials-16-02033-f016:**
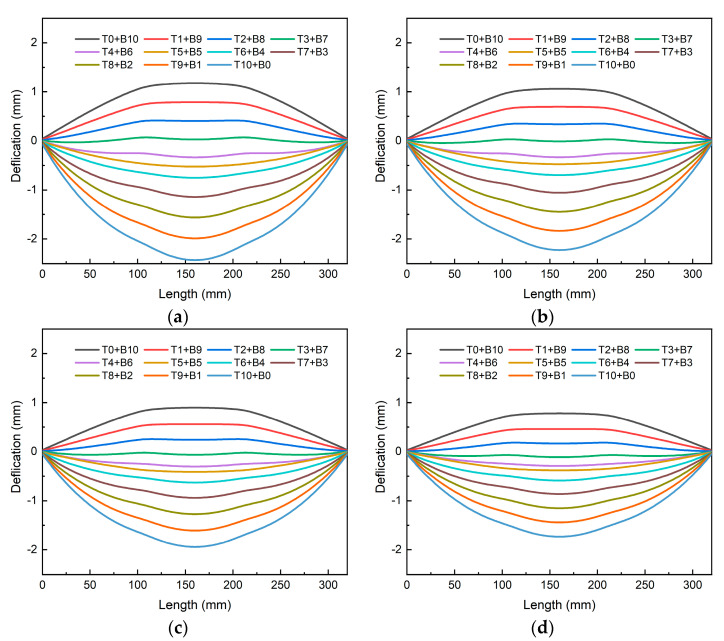
The bottom deformation of parts under different machining strategies with the different initial stress states: (**a**) 1:1; (**b**) 1:0.75; (**c**) 1:0.5; (**d**) 1:0.25.

**Figure 17 materials-16-02033-f017:**
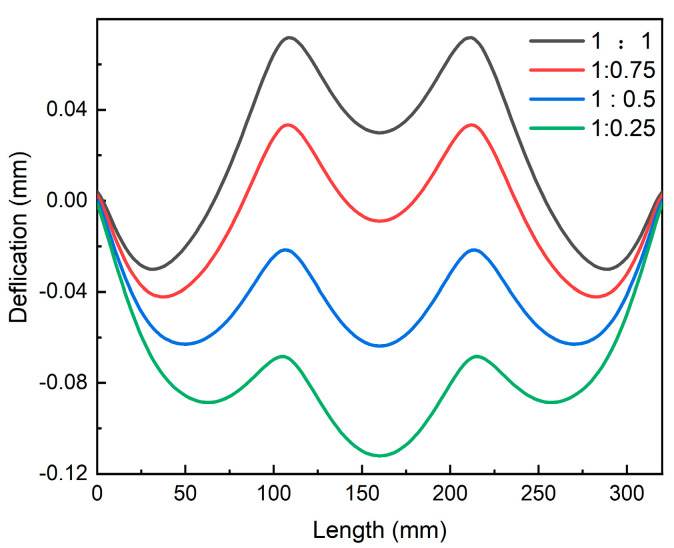
The bottom deformation of parts under T3+B7 machining strategies.

**Figure 18 materials-16-02033-f018:**
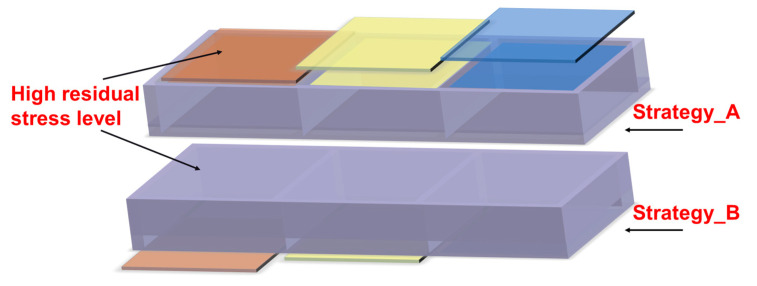
Diagram of the machining techniques.

**Figure 19 materials-16-02033-f019:**
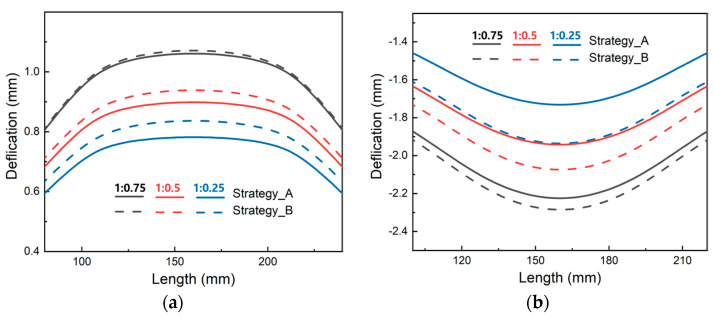
The bottom deformation of components using various machining techniques with various initial stress states: (**a**) T0+B10; (**b**) T10+B0.

**Table 1 materials-16-02033-t001:** Mechanical and physical properties of aluminum alloy 7055.

Temperature (°C)	20	100	200	300	400	500
ρ (kg·m^−3^)	2841	2828	2807	2787	2761	2735
λ (W·m^−1^·°C^−1^)	126.7	133.3	179.4	179.4	191.5	175.4
α (10^−6^·°C^−1^)	22.7	24.0	24.2	25.2	26.0	27.5
c (J·kg^−1^∙K^−1^)	913	983	1025	1113	1292	1158
*R_p0.2_* (MPa)	266	223	154	73	23	10
*E* (GPa)	73	61	56	38	32	25

**Table 2 materials-16-02033-t002:** Heat transfer coefficient of different spray flow rates.

$Water\;flow\;rate\;(m^{3}\cdot h^{-1})$	0.64	0.48	0.32	0.16
$Heat\;transfer\;coefficients\;(W\cdot m^{-2}\cdot {{}^{\circ}C}^{-1})$	10,000	8000	5500	4000

**Table 3 materials-16-02033-t003:** Machining parameters used in the machining experiments.

Rotation Speed (r·min^−1^)	1200
Feed speed (mm·min^−1^)	150
Axial cutting depth (mm)	1
Radial cutting width (mm)	4
Tool inclination angle	30°
Tool cearance angle	8°

## Data Availability

Not applicable.

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
