# Peer review of "Influence of Material Removal Strategy on Machining Deformation of Aluminum Plates with Asymmetric Residual Stresses"

_materials, 2023, doi:10.3390/ma16052033_

Round 1

Reviewer 1 Report

I did not find the manuscript and its results very interesting and scientifically sound. The methodology of result presentation must be improved.

Reviewer 2 Report

The submitted study analyzes the influence of material removal strategy on machining deformation of aluminum thick plates by a combination of finite element simulation and experiments. 

The experiments can show that with a reduction in the spray flow rate following spray quenching, the stress level can be reduced. Furthermore, the authors report, that the machining distortion of the plates can be decreased by using an adapted material removal approach. Additionally, the lower the stress level of the plate, the smaller the final deformation.

The experiments are well performed, points that could be additionally addressed are how rotational speed and inclination angle could affect the analysis parameters. Other variables could also have an impact on the deformation and should be addressed as a limitation in the discussion section.

Reviewer 3 Report

The authors investigated the Influence of material removal strategy on machining deformation of aluminum plates with asymmetric residual stresses. The following are my suggestions.

1. Remove the word thick from the manuscript title.

2. Abstract: T3+B7 and T10+B0 could not be understood. Explain initially (Tm+Bn).

3. Provide some numarical findings in the abstract.

4. Avoid starting the sentences like, Finally, Additionally etc. throughout the manuscript.

5. Provide reference for "Birth and Death" process in line 123.

6. The resolution of all the figures have to be improved >300dpi. Many are poor and are blurred.

7. Define the novelty of the manuscript in comparison to your previous studies. Also include future work.

8. How much deviation obtained between experimental and simulation?

9. Check and prepare the manuscript as per mdpi format including reference. There are some minor mistakes.

Reviewer 4 Report

Some of the figures need higher quality images and annotations like Figure 14.

Figure 15, the font for the colour map is not clear.

Section 2.1.2: please include the undeformed mesh image and element type used.

Reviewer 5 Report

Influence of material removal strategy on machining deformation of aluminum thick plates with asymmetric residual stresses

Review Report

1. What is the main question addressed by the research?

In this the effects of material removal strategies and initial stress states on the machining deformation of aluminum alloy plates were investigated through a combination of finite element simulation and experiments.

The asymmetric initial stress state has a significant impact on the machining deformation of the thick plate. 16 The machined deformation of thick plates increases with the increase of the initial stress state.

2. Do you consider the topic original or relevant in the field? Does it

address a specific gap in the field?

The topic is relevant in the field because asymmetric distribution of residual stress of the thick plates were implemented by means of finite element simulation and asymmetric spraying experiments. Many authors in field are still dedicating research time to understanding the impact of initial residual stress state on machining deformation in the milling of monolithic aircraft components and finding ways to minimize the final distortion.

3. What does it add to the subject area compared with other published

material?

New information in the field, concerning the machining deformation of aluminum thick plates with asymmetric residual stresses

4. What specific improvements should the authors consider regarding the

methodology? What further controls should be considered?

 The informations in the manuscript are okay presented.

5. Are the conclusions consistent with the evidence and arguments presented

and do they address the main question posed?

  Yes.

6. Are the references appropriate?

 Yes.

7. Please include any additional comments on the tables and figures.

 English and sentences length should be re-checked.
